

# Geographic population genetic structure and diversity of *Sophora moorcroftiana* based on genotyping-by-sequencing (GBS)

Ying Liu[1,2], Fei Yi[1], Guijuan Yang[1], Yuting Wang[3], Ciren Pubu[3], Runhua He[4], Yao Xiao[1], Junchen Wang[1,2], Nan Lu[1], Junhui Wang[1] and Wenjun Ma[1]

[1] State Key Laboratory of Tree Genetics and Breeding, Key Laboratory of Tree Breeding and Cultivation of State Forestry Administration, Research Institute of Forestry, Chinese Academy of Forestry, Beijing, China
[2] College of Forestry, Northwest A&F University, Yangling, Shaanxi, China
[3] Forest Science Research Institute of Tibet Municipality, Lhasa, Tibet, China
[4] College of Forestry, Central South University of Forestry and Technology, Changsha, Hunan, China

Corresponding author
Wenjun Ma, mwjlx.163@163.com

## ABSTRACT

*Sophora moorcroftiana* is a perennial leguminous low shrub endemic to the Yarlung Zangbo River basin in Tibet with irreplaceable economic and ecological value. To determine the drivers of evolution in this species, 225 individuals belonging to 15 populations from different geographic locations were sampled, and population genetics was studied using high-throughput genotyping-by-sequencing (GBS). Based on genetic diversity analysis, phylogenetic analysis, principal component analysis, and structure analysis, 15 natural populations were clustered into the following five subgroups: subgroup I (Shigatse subgroup) was located in the upper reaches of the Yarlung Zangbo River with a relatively high level of population genetic variation (means for PIC, Shannon and PI were 0.173, 0.326 and 0.0000305, respectively), and gene flow within the subgroup was also high (mean value for *Nm* was 4.67). Subgroup II (including Pop 7 and Pop 8; means for PIC, Shannon and PI were 0.182, 0.345 and 0.0000321, respectively), located in the middle reaches of the Yarlung Zangbo River had relatively high levels of gene flow with the populations distributed in the upper and lower reaches. The *Nm* between subgroup II with subgroups I and III was 3.271 and 2.894, respectively. Considering all the genetic diversity indices Pop 8 had relatively high genetic diversity. Subgroup III (the remaining mixed subgroup of Lhasa and Shannan) was located in the middle reaches of the Yarlung Zangbo River and the means for PIC, Shannon and PI were 0.172, 0.324 and 0.0000303, respectively. Subgroup IV (Nyingchi subgroup), located in the lower reaches of the Yarlung Zangbo River basin, showed a further genetic distance from the other subgroups and the means for PIC, Shannon and PI were 0.147, 0.277 and 0.0000263, respectively. Subgroup V (Nyingchi Gongbu Jiangda subgroup), located in the upper reaches of the Niyang River, had the lowest level of genetic variation (means for PIC, Shannon and PI were 0.106, 0.198 and 0.0000187, respectively) and gene flow with other populations (mean value for *Nm* was 0.42). According to the comprehensive analysis, the *S. moorcroftiana* populations generally expanded from upstream to downstream and displayed a high level of genetic differentiation in the populations in the upper and lower reaches. There were high levels of gene exchange between the central populations with upstream and downstream populations, and

wind-induced seed dispersal was an important factor in the formation of this gene exchange mode.

## INTRODUCTION

Molecular markers are extremely useful in plant and animal genetics and genomics. Single nucleotide polymorphism (SNP) genetic markers are one of the key tools for population genetics and quantitative genetics. Genotyping-by-sequencing (GBS) is a high-throughput, multiplex and short-read sequencing approach that reduces genome complexity via restriction enzymes and generates high-density genome-wide markers at a low cost per sample by tagging random DNA fragments shared by different samples with unique, short DNA sequences (barcodes) and pooling samples into a single sequencing channel (*Elshire et al., 2011*). This approach not only greatly reduces the cost of sequencing but also makes it possible to genotype large samples. GBS is of great significance for understanding the genetic background and phylogeny of germplasm resources (*Poland et al., 2012a*; *Poland et al., 2012b*; *Poland & Rife, 2012c*; *Glaubitz et al., 2014*). The short-read sequences obtained by GBS can be assembled by a reference genome or nonreference genome to obtain high-density SNP markers. These SNP markers or developed bin markers can be used for various processes, such as genetic map construction (*Ward et al., 2013*), genome-wide association studies (GWAS) (*Sakiroglu & Brummer, 2017*), and genome assembly. At present, GBS technology is an important tool for genotyping and is widely used in genetic linkage mapping, genetic selection (*Zhang et al., 2018a*; *Zhang et al., 2018b*), genetic diversity studies (*Lu et al., 2013*), germplasm identification (*Wu et al., 2014*), species identification (*Pembleton et al., 2016*) and other fields.

The Yarlung Zangbo River basin stretches over large areas of the Lhasa, Shigatse and Shannan regions of Tibet and over small areas within the Ngari, Nagqu and Qamdo regions, including 41 counties (cities). The relief of the basin is high in the west, low in the east, high in the north and south, and low in the central reaches. The Yarlung Zangbo River basin is sensitive to the evolution of the eco-environment, as most of the ground surface of the basin is below the lower one-third of the air convection layer and is therefore vulnerable to global climate change. Due to the monsoons and subtropical westerly jet over the plateau, the river valley exhibits a dry, cold and windy climate (*Dong et al., 1997*). As the orientation of the river valley is nearly parallel to the wind direction, the mountainous terrain greatly increases the wind velocity. Thus, the Yarlung Zangbo River valley has favorable environmental conditions for the development of aeolian sand landforms, including a sand source, a wind driving force, and deposition fields (*Li et al., 1999*). In total, there were 273,697.54 ha of aeolian sandy land in the Yarlung Zangbo River basin in 2008 (*Shen et al., 2012*). Desertification in Tibet is an urgent problem that must be addressed (*Zou et al., 2002*). *Sophora moorcroftiana* is a vital species for desertification control.

*S. moorcroftiana* (Benth.) Benth. *ex* Baker (Fabaceae), a diploid (2n =2x =18; (*Wang, Liu & Xu, 1995*), is acknowledged as a long-lived perennial shrub species endemic to the Qinghai-Tibet Plateau (QTP). This species exists in China Tibet as well as Bhutan, N Burma, and N India (*Liu et al., 2006*) and is mainly distributed in the middle and upper reaches of the dry valley region of the Yarlung Zangbo River in Tibet with altitudes ranging from 2,800 m to 4,400 m (*Li et al., 2017*). It plays an irreplaceable role in Tibet as important forage, and the seeds are used in China as a crude drug (*Chang, 1977*). Furthermore, the species is currently the preferred drought-resistant afforestation tree species due to its strong adaptability to sand burial in the plateau (*Zhao, Zhang & Li, 2007*), leading to its irreplaceable role in vegetation reestablishment programs aiming to stabilize shifting sand (*Liu et al., 1998*). Meanwhile, the species has received much attention due to its medicinal properties (*Zhang et al., 2018a*; *Zhang et al., 2018b*).

Another *Sophora* species in the QTP, *S. davidii*, which is an important leguminous shrub widely distributed in southeastern China (*Zhao et al., 2016*), is closely related to *S. moorcroftiana* (*Wu, 1983*) and is widely distributed from the southeast of the QTP to central China. Although these two species share many characteristics, such as being diploid (2n = 18), insect-pollinated, and gravity-dispersed via propagules, a hypothesis was proposed in which *S. moorcroftiana* diverged from *S. davidii* and speciated (*Shen, 1996*). Many studies have found large differences in the phenotypes of *S. moorcroftiana* at different altitudes in the Yarlung Zangbo River basin, but few studies focus on its genetics in different locations. Understanding the genetic evolution of *S. moorcroftiana* has important implications for germplasm conservation and vegetation reconstruction. Even though *Liu et al. (2006)* has reported the genetic variation in *S. moorcroftiana* examined using allozyme markers, existing research in this area is still rare. To explore the relationships between the distribution and expansion of *S. moorcroftiana* and geographical factors, we employed GBS to call SNPs of 15 *S. moorcroftiana* populations originated from different geographical locations and leveraged them for further analyses.

The specific goals of this study are (1) to determine the genetic characteristics within and among populations; (2) to examine the effects of altitude and longitude on the population genetic characteristics; and (3) to find historical, life history, and/or environmental factors that might explain the patterns and levels of observed genetic variation.

## MATERIALS & METHODS

### Plant material collection

To determine the correlations between the evolution of *S. moorcroftiana* populations and geographical factors, we utilized the following sampling design: (1) we selected sampling points along a river but separated by a mountain (Pop 8 and Pop 9, Lhasa River Basin) to verify the relationship between the population expansion and water flow; (2) we selected sampling sites from the highest (Pop 3, Shigatse) altitude to the lowest (Pop 14, Nyingchi) altitude and from the lowest longitude (Pop 1, Shigatse) to the maximum longitude (Pop 15, Nyingchi) where *S. moorcroftiana* exists to verify the impacts of altitude and longitude on the population genetic characteristics; and (3) we selected sampling sites on different

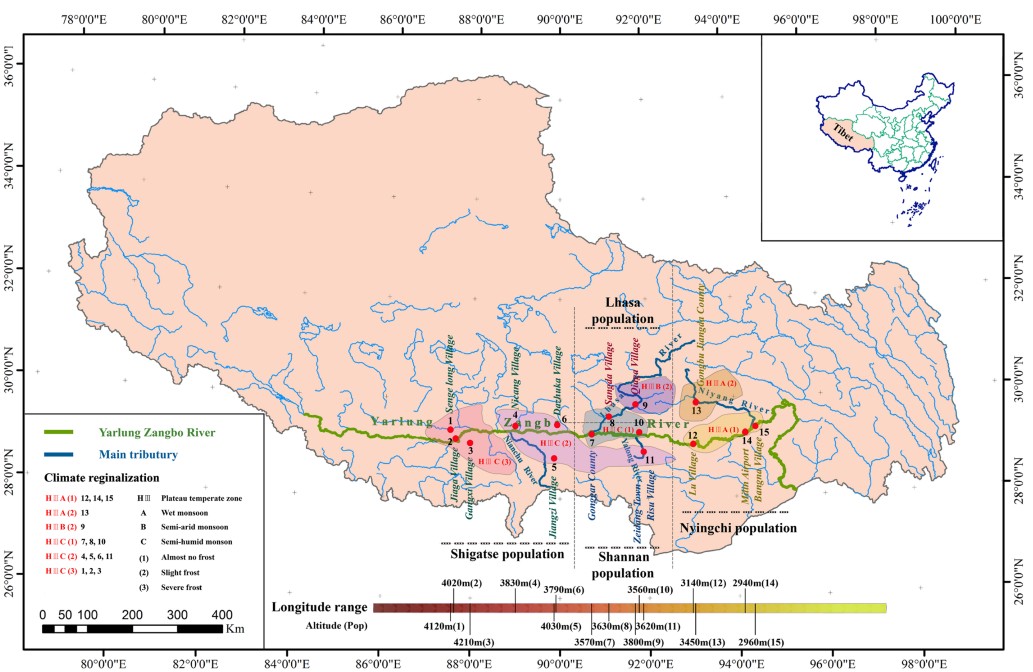

**Figure 1** **Geographic distribution of sampled *S. moorcroftiana* populations in the Yarlung Zangbo River basin.** Each population is indicated by a dot on the map. The information of climate regionalization information was obtained from *Yang (2013)*.

slopes (e.g., Pop 7 and Pop 8, Pop 9 and Pop 10, and Pop 12 and Pop 13) and in different directions to verify the relationship between the populations and the direction of sand flow caused by the slope and wind (Fig. 1). We obtained a total of 225 samples representing 15 natural populations from 15 locations (15 individuals per location), which were used for the analysis (Table 1), and the populations were numbered according to longitude. In addition, *Sophora davidii* (Franch.) (individuals 226–229) was used as the outgroup to prevent ascertainment bias. For DNA extraction, young leaf tissue was collected, immediately frozen in liquid nitrogen, and stored at −80 °C.

## DNA extraction and GBS

The DNA of the young leaves of *S. moorcroftiana* and *S. davidii* individuals was extracted for GBS using a modified cetyltrimethylammonium bromide (CTAB) extraction method (*Tel-Zur et al., 1999*). We employed a two-enzyme (MseI and TaqaI) GBS protocol modified from a previously described protocol (*Poland et al., 2012a*; *Poland et al., 2012b*) to generate a library consisting of DNA fragments with a barcode and performed sequencing with an Illumina HiSeq PE 150. The average sequencing depth for each sample was 10×.

## SNP data analysis

Quality control of the FASTQ-format raw data was performed with the FastQC application (*Brown, Pirrung & Mccue, 2017*) to ensure that there were no hidden problems. Adapter sequences and abnormal nucleotide bases at the 5′terminus were removed from the raw sequencing reads using FastQC 0.6.0. Additionally, the low

Liu et al. (2020), *PeerJ*, DOI 10.7717/peerj.9609

**Table 1  Geographic and climate information for the *S. moorcroftiana* populations in the study.**  All the climate information was obtained from *Yang (2013)*.

| Population number | Individual number | Origin | Altitude (m) | Longitude (E°) | Latitude (N°) | Mean annual temperature (°C) | Mean annual precipitation (mm) | Mean annual wind velocity (m/s) |
|---|---|---|---|---|---|---|---|---|
| Pop 1 | 61–75 | Shigatse, Senge long Village | 4120 | 87°31′49″ | 29°09′24″ | 5 | 400 | 2.6~3.0 |
| Pop 2 | 76–90 | Shigatse, Chawu Township, JiagaVillage | 4020 | 87°32′21″ | 29°06′57″ | 5 | 400 | 2.6~3.0 |
| Pop 3 | 46–60 | Shigatse, Gangxi Village | 4210 | 87°55′57″ | 29°03′40″ | 5 | 400 | 2.6~3.0 |
| Pop 4 | 211–225 | Shigatse, Nicang Village | 3830 | 88°55′80″ | 29°19′00″ | 5 | 400 | 1.0~1.4 |
| Pop 5 | 196–210 | Shigatse, Near Jiangzi County | 4030 | 89°36′18″ | 28°54′51″ | 5 | 500 | 1.8~2.2 |
| Pop 6 | 106–120 | Shigatse, West of Dazhuka | 3790 | 89°37′17″ | 29°20′28″ | 5 | 400 | 1.4~1.8 |
| Pop 7 | 16–30 | Shannan, Gonggar County | 3570 | 90°52′15″ | 29°17′38″ | 7 | 400 | 1.8~2.2 |
| Pop 8 | 1–15 | Lhasa, Sangda Village | 3630 | 91°01′49″ | 29°33′26″ | 5 | 500 | 1.8~2.2 |
| Pop 9 | 166–180 | Lhasa, Mozhu, Gongka Town, Qiaga Village | 3800 | 91°43′41″ | 29°50′10″ | 3 | 500 | 1.8~2.2 |
| Pop 10 | 151–165 | Shannan, North of Zeidang Town | 3560 | 91°47′09″ | 29°14′42″ | 7 | 400 | 2.2~2.6 |
| Pop 11 | 136–150 | Shannan, Naidong County, Risu Village | 3620 | 91°52′29″ | 29°00′33″ | 7 | 400 | 2.2~2.6 |
| Pop 12 | 91–105 | Nyingchi, west of Lang County, Lu Village | 3140 | 93°02′31″ | 29°02′57″ | 7 | 500 | 1.8~2.2 |
| Pop13 | 181–195 | Nyingchi, Gongbu Jiangda County | 3450 | 93°11′30″ | 29°53′50″ | 3 | 500 | 1.8~2.2 |
| Pop 14 | 121–135 | Nyingchi, Near Milin Airport, Hongwei Forest | 2940 | 94°20′10″ | 29°19′06″ | 7 | 700 | 1.8~2.2 |
| Pop 15 | 31–45 | Nyingchi, Bangna Village | 2960 | 94°27′36″ | 29°27′27″ | 7 | 800 | 1.8~2.2 |

quality ends of the reads (sequence quality < Q 20) were trimmed and we removed reads that contained 10% Ns (undefined bases) and sequence lengths less than 25 bp after trimming. Then, the bash command cat was used to combine the two sequences generated by paired-end sequencing of each sample into one sequence. Without a reference sequence, SNP calling for each sample was performed using the Stacks (http://catchenlab.life.illinois.edu/stacks/source/stacks-2.41.tar.gz 2.0 Beta 8, http://catchenlab.life.illinois.edu/stacks/) pipeline to build loci (ustacks), create a catalog of loci (cstacks), match samples back to the catalog (sstacks), transpose the data (tsv2bam), add paired-end reads to the analysis, call genotypes, and perform population genomics analysis (*Catchen et al., 2013*). The following SNP-filtering parameters were used to construct the phylogenetic tree in VCFtools v4.0 (*Danecek et al., 2011* http://vcftools.github.io/): -min DP 5 (minor depth $\geq$ 5), -max-missing 0.8, -maf 0.05 (minor allele frequencies $\geq$ 5%) and a distance between variants of 300 bp. A phylogenetic tree of 229 samples was constructed using the neighbor-joining (NJ) method by Mega7.0 in Linux. The tree file was imported into the Interactive Tree of Life (ITOL) (https://itol.embl.de/itol.cgi) (*Letunic & Bork, 2016*), which is an online tool for the display, annotation and management of phylogenetic trees. ADMIXTURE linux-1.3.0 (*Alexander, Novembre & Lange, 2009*) is a software that is commonly used for population structure analysis. Bayesian model-based clustering was used to assign all individuals from all geographic locations to taxa by ADMIXTURE. The number of genetic clusters, $K$, was tested from 2 to 15, and 14 independent runs were computed for each $K$ value. Principal component analysis (PCA) is a pure mathematics method that can select a small number of important variables from multiple related variables. PCA of SNPs obtained from the 229 samples was performed using GCTA (*Yang et al., 2011*) (http://cnsgenomics.com/software/gcta/#PCA) software for the clustering of the main components. The nucleotide diversity (PI, *Nei, 1987*), Tajima's D (*Tajima, 1989*), Shannon–Wiener index (*Keylock, 2005*), polymorphic information content (PIC, *Nagy et al., 2012*), observed heterozygosity ($H$o), and the expected heterozygosity (*Berg & Hamrick, 1997*) of the 15 *S. moorcroftiana* populations and the fixation index (*Weir & Cockerham, 1984*) among populations were calculated by Stacks. Linear regression analysis of the PIC, Shannon–Wiener index and PI with altitude and longitude was conducted using Excel. By inverting Wright's formula (*Wright, 1951*), the value of $Nm$ can be estimated from $F_{ST}$, as $Nm \approx (1- F_{ST})/ 4 F_{ST}$, where N is the effective population size of each population and m is the migration rate between populations. This method is effective for estimating gene flow indirectly.

## RESULTS

### Phylogenetic analysis

In the phylogenetic analysis of the 229 samples (Fig. 2), *S. davidii* individuals 226–229 (outgroup) were the farthest from the other individuals and separated from other individuals at the base of the tree. Within the *S. moorcroftiana* populations, Pop 13 (Nyingchi, Gongbu Jiangda county), which contained individuals 181–195, was the first population that separated from the other populations. Pop 12 (individuals 91–105), Pop

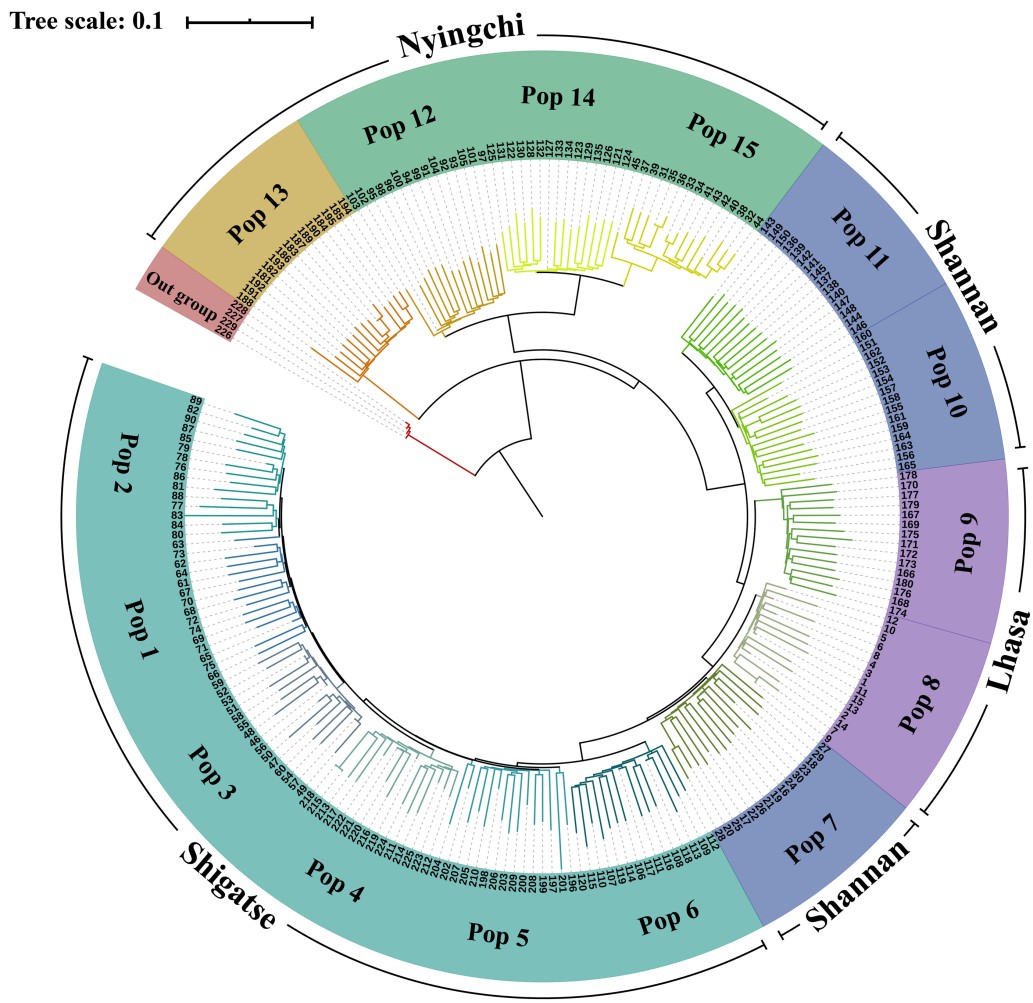

**Figure 2** **Neighbor-Joining tree based on SNP data of 225 _S. moorcroftiana_ individuals and four _S. davidii_ individuals.** The periphery of the tree includes the population location information and population information, and the inner part indicates the individual number.

14 (individuals 121–135) and Pop 15 (individuals 31–45) from Nyingchi in the Yarlung Zangbo River basin were relatively close and separated from the other populations. Pop 11 (individuals 136–150) and Pop 10 (individuals 151–165) were clustered together and were relatively far from the other populations. Compared to Pop 8 (individuals 1–15) and Pop 7 (individuals 16–30), Pop 9 (individuals 166-180) in the upper reaches of the same river basin (Lhasa River) was far away. Among the Shigatse populations, Pop 6 (individuals 106–120) separated from the other populations first (including Pop 5, Pop 4, Pop 3, Pop 1, and Pop 2) and then Pop 5 (individuals 196–210) separated. Meanwhile, Pop 1 was close to Pop 2 and Pop 3 was close to Pop 4. Overall, the populations located in Nyingchi separated first from the populations in other areas, followed by the Shannan populations, Lhasa populations, and finally the Shigatse populations, and there is east–west differentiation.

## Population structure analysis

Estimated ancestries, derived from multilocus genotype data, can be used to perform statistical correction for population stratification (*Alexander, Novembre & Lange, 2009*). We used ADMIXTURE, which adopts the likelihood model embedded in STRUCTURE and runs considerably faster than other tools to estimate ancestry, to identify possible subgroups. When $K = 5, 6, 7$ and 8, the cross validation errors (CV errors) were relatively low, with $K = 7$ minimizing the CV error (Fig. 3). Clustering information for the 225 *S. moorcroftiana* individuals when $K = 2$ to 9 is shown in Fig. 3B. Moving from $K = 2$ to 9, the large samples from the upper and middle reaches of the Yarlung Zangbo River were first distinguished from the lower and middle reaches, after which the samples from the middle reaches (Shannan, Lhasa) were revealed as distinct. When all the samples were clustered into four taxa, the genetic information of Pop 1, Pop 2, Pop 3, Pop 4, Pop 5 and Pop 6 from the upper reaches of the Yarlung Zangbo River was derived from an ancestor. Pop 9, Pop 10, and Pop 11 were derived from another ancestor, while approximately half of the genetic information for Pop 7 and Pop 8 came from the ancestor of Pop 1–6 and the remaining half came from the ancestor of Pop 9–11. Pop 14 and Pop 15 were derived from an ancestor and approximately 70% of the information for Pop 12 was from the ancestor of Pop 14, Pop 15. Pop 13 was derived from a single ancestor. When $K = 5$, Pop 9 was divided into a different taxon with Pop 10 and Pop 11, and the genetic information for Pop 7 and Pop 8 was still derived from two ancestors. However, the genes were more clearly identified as being from Pop 9. When $K = 6$, Pop 7 and Pop 8 were divided into a different taxon with Pop 9. A portion of the genetic information for Pop 12 was from the ancestor of Pop 10 and Pop 11 and the ancestor of Pop 13, which indicates that genetic information can be extended over a certain distance. When $K = 7$, Pop 12 was divided into different taxa with Pop 14 and Pop 15, and Pop 14 contained some genetic information from Pop 12.

## Principal component analysis

To supplement the structural analysis results, we used GCTA for PCA. The PCA of SNPs clearly distinguished five major subgroups among the 225 *S. moorcroftiana* individuals (Figs. 4A–4C), with the outgroup containing individuals 226–229 excluded. This result confirmed the population structure analysis results when $K = 6$, but the only difference is that Pop 9, Pop 10 and Pop 11 were clustered into the same subgroup by PCA. Subgroup I was in the high-altitude and westernmost region of the sampling area, which contained Pop 1, Pop 2, Pop 3, Pop 4, Pop 5, and Pop 6 from Shigatse (Fig. 4D). Subgroup II contained Pop 7 from Shannan and Pop 8 from Lhasa. Pop 7 was from the confluence of the Lhasa River and the Yarlung Zangbo River and Pop 8 belonged to the Lhasa River Basin in detail. Subgroup III was in the middle of the sampling area (Lhasa and Shannan) and contained Pop 9, Pop 10, and Pop 11. Specifically, Pop 9 belonged to the Lhasa River Basin, Pop 10 was from the confluence of the Yalong River and the Yarlung Zangbo River, and Pop 11 belonged to the Yalong River basin. Subgroup IV was in the lowest and easternmost region of the sampling area (Nyingchi, Yarlung Zangbo River basin) and contained Pop 12, Pop 14, and Pop 15. Among these populations, Pop 15 was from the lower reaches of

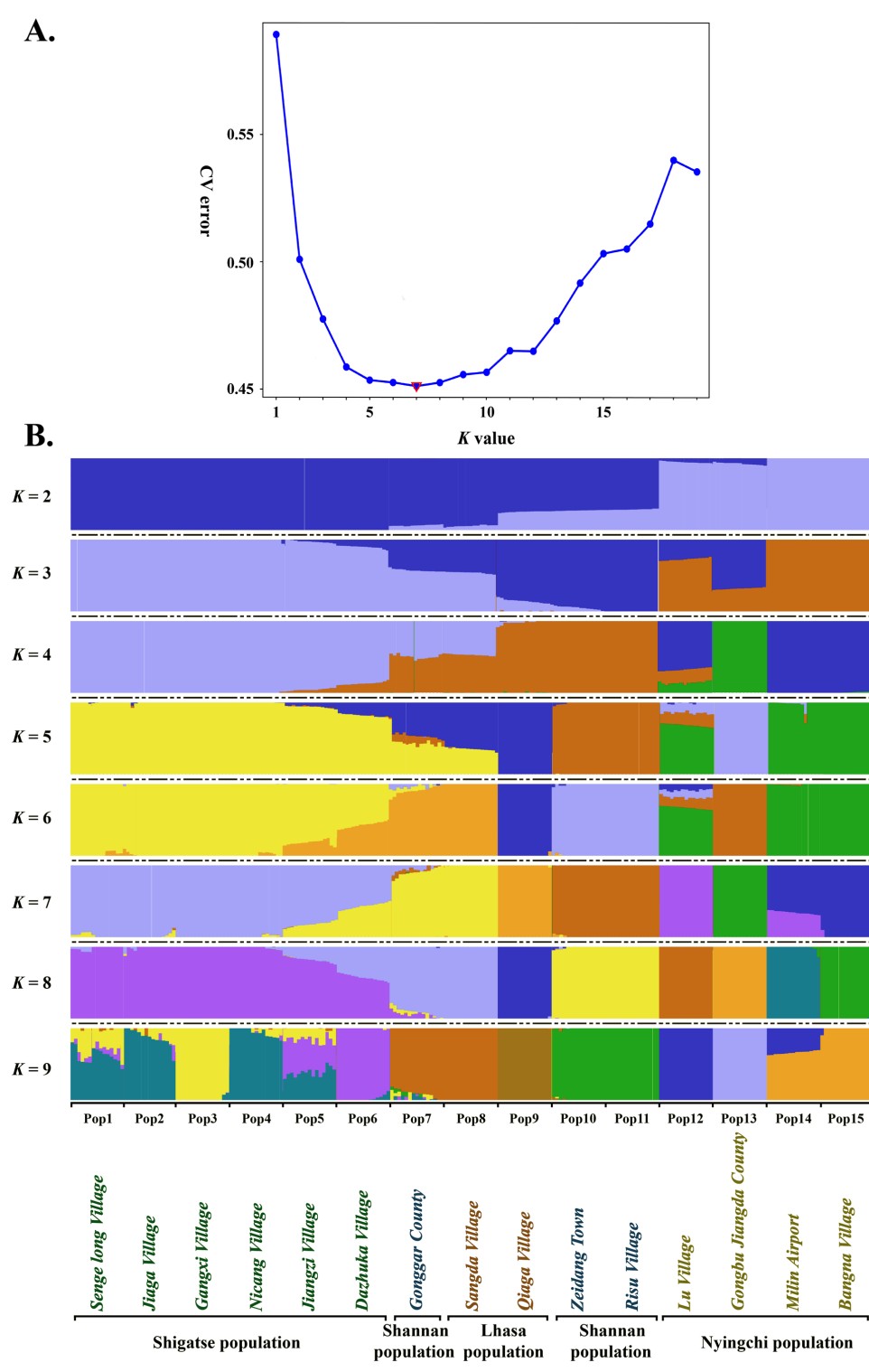

**Figure 3 Population genetic structure analysis based on SNP data of 225 *S. moorcroftiana* individuals (*K* = 2 to 9).** (A) The CV error varies among *K* values. (B) Clustering information for different individuals when *K* = 2 to 9.

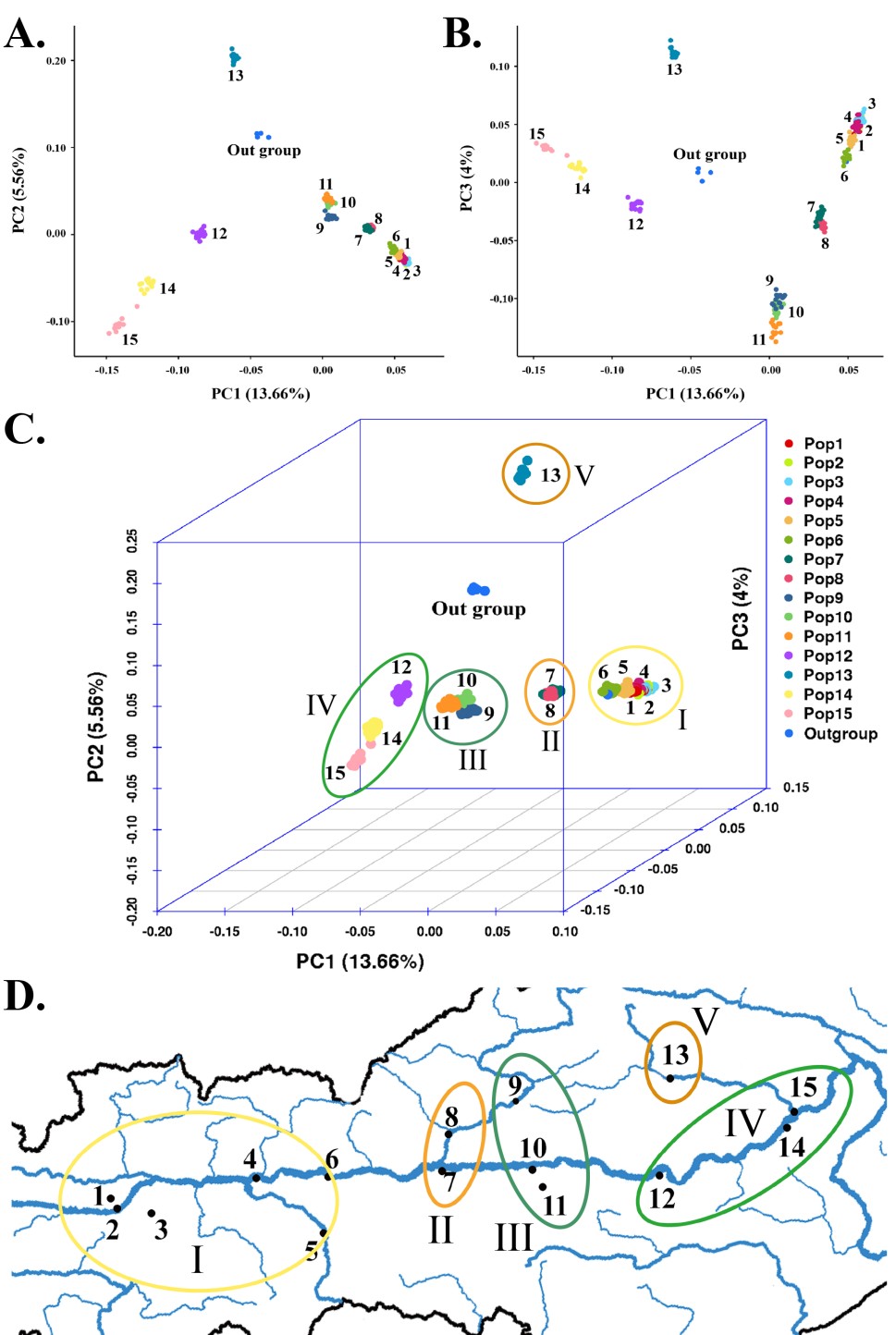

**Figure 4** **Principal component analysis based on SNP data of 225 *S. moorcroftiana* individuals and 4 *S. davidii* individuals.** (A) PC1 & PC2. (B) PC1 & PC3. (C) PCA in 3D. (D) PCA results corresponding to the geographical locations.

the Niyang River. Subgroup V was located at high altitudes in the eastern sampling area (Nyingchi, Gongbu Jiangda county) and only contained Pop 13, which was from the upper reaches of the Niyang River. The analyses showed that Pop 13 was obviously separated and differentiated from the other populations. The clustering of populations by PCA roughly conformed to the geographic distribution of the populations.

## Population genetic diversity

Relatively low levels of genetic diversity were found in the *S. moorcroftiana* populations (Table 2). The PIC is an indicator of the level of polymorphism. The PIC was between 0 and 1 among the 15 populations, with low polymorphism (PIC < 0.25) among populations. The PIC ranged from 0.106 (Pop 13) to 0.183 (Pop 8), with an average of 0.164. The Shannon–Weiner index incorporates the following two factors: species richness and the equitability or evenness of individual distributions. Among all the populations, the index ranged from 0.198 (Pop 13) to 0.346 (Pop 8). PI is a concept used in molecular genetics to measure the degree of polymorphism within a population. The PI ranged from 0.00001873 (Pop 13) to 0.00003233 (Pop 8), with an average of 0.00002904. The PIC, Shannon, and PI levels of the 15 populations decreased with increasing longitude (Figs. 5A–5C) and increased with increasing altitude (Fig. 5D). Although these indices were negatively correlated with longitude overall, populations at an intermediate longitude, especially Pop 7 and Pop 8, had the highest PIC, Shannon, and PI levels. The longitude of Pop 13 was not the highest, but the PIC, Shannon, and PI levels were the lowest. Tajima's D is a statistical test that is mainly used to distinguish whether a DNA sequence has undergone random or nonrandom evolution, and whether or not it underwent directional selection. $T = 0$ indicates that no selection occurred. Tajima's D was greater than 0 for all the populations, ranging from 0.712 (Pop 10) to 1.296 (Pop 13). The observed heterozygosity ($Ho$) ranged from 0.2701 (Pop 11) to 0.3249 (Pop 8) and the expected heterozygosity (*Berg & Hamrick, 1997*) ranged from 0.2956 (Pop 7) to 0.3382 (Pop 14). Taking all indicators into account, Pop 8 had relatively high genetic diversity.

## Population genetic differentiation

$F_{ST}$ is a measure of genetic differentiation among populations, and studying the genetic structure of a population is essential to understanding its evolutionary properties (*Whitlock & Mccauley, 1999*). Wright (*Wright, 1965*) suggested that an $F_{ST}$ ranging from 0 to 0.05 indicates very little genetic differentiation between populations and is thus not worth considering; $F_{ST}$ ranging from 0.05 to 0.15 indicates moderate genetic differentiation between populations; $F_{ST}$ ranging from 0.15 to 0.25 indicates large genetic differentiation between populations; and $F_{ST}$ greater than 0.25 indicates strong genetic differentiation between populations. As shown in Table 3, the $F_{ST}$ between pairs of the 15 populations ranged from 0.021 (Pop 1 with Pop 2) to 0.458 (Pop 15 with Pop 13), with an average of 0.222, which indicated large genetic differentiation among *S. moorcroftiana* populations. Within the Shigatse populations, Pop 1 had very little genetic differentiation from Pop 2, Pop 5, Pop 3 and Pop 4; Pop 2 had very little genetic differentiation from Pop 5; and Pop 5 was close to Pop 6. Overall, the genetic relationship of the populations in Shigatse

**Table 2  Estimates of genetic variability of 15 *S. moorcroftiana* populations.**

| Population | PIC | Shannon | PI | Tajimas'D | *H*o | *H*e |
|---|---|---|---|---|---|---|
| Pop 1 | 0.175 | 0.331 | 0.00003088 | 0.829 | 0.2892 | 0.3018 |
| Pop 2 | 0.171 | 0.323 | 0.00003019 | 0.838 | 0.2724 | 0.3023 |
| Pop 3 | 0.168 | 0.317 | 0.00002975 | 0.962 | 0.3223 | 0.3098 |
| Pop 4 | 0.169 | 0.319 | 0.00002988 | 0.884 | 0.3193 | 0.3048 |
| Pop 5 | 0.178 | 0.336 | 0.00003135 | 0.787 | 0.2768 | 0.3014 |
| Pop 6 | 0.175 | 0.332 | 0.00003091 | 0.755 | 0.2878 | 0.2971 |
| Pop 7 | 0.181 | 0.343 | 0.00003188 | 0.734 | 0.2728 | 0.2956 |
| Pop 8 | 0.183 | 0.346 | 0.00003233 | 0.877 | 0.3249 | 0.3048 |
| Pop 9 | 0.168 | 0.317 | 0.00002972 | 0.900 | 0.2821 | 0.3060 |
| Pop 10 | 0.175 | 0.331 | 0.00003080 | 0.712 | 0.2709 | 0.2965 |
| Pop 11 | 0.172 | 0.325 | 0.00003031 | 0.831 | 0.2701 | 0.3019 |
| Pop 12 | 0.159 | 0.300 | 0.00002834 | 1.046 | 0.2931 | 0.3170 |
| Pop 13 | 0.106 | 0.198 | 0.00001873 | 1.296 | 0.2933 | 0.3364 |
| Pop 14 | 0.155 | 0.290 | 0.00002769 | 1.411 | 0.3147 | 0.3382 |
| Pop 15 | 0.128 | 0.241 | 0.00002291 | 1.279 | 0.3168 | 0.3296 |
| Mean (at population level) | 0.164 | 0.310 | 0.00002904 | 0.943 | 0.2938 | 0.3095 |
| Species-level values | | | | | 0.1919 | 0.2661 |

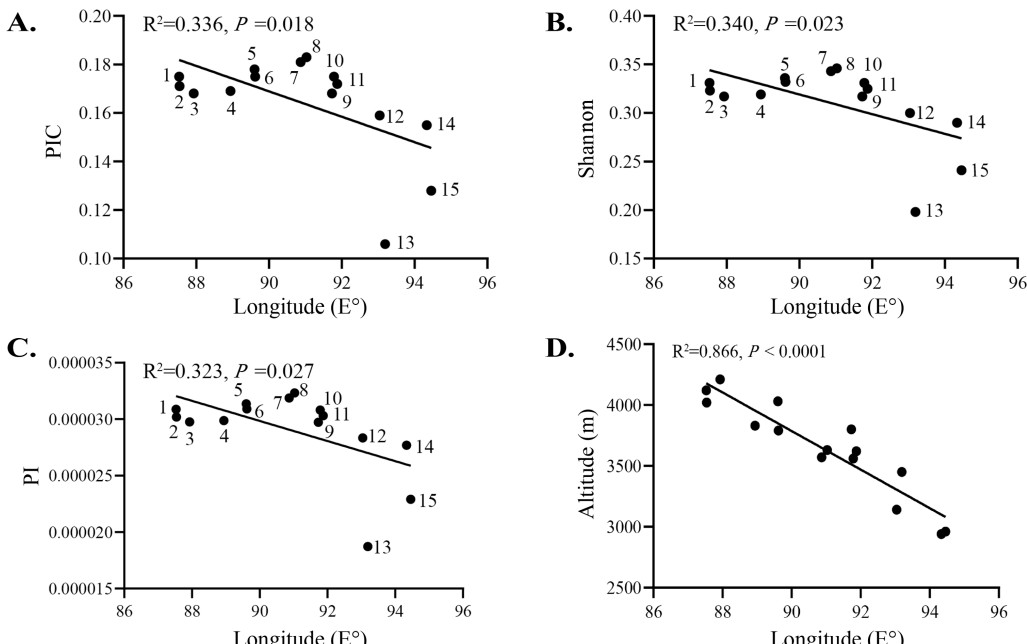

**Figure 5  Relationship between genetic variation of 15 *S. moorcroftiana* populations and longitude.**
(A) Relationship between PIC and longitude. (B) Relationship between Shannon and longitude. (C) Relationship between PI and longitude. (D) Relationship between altitude and longitude.

**Table 3  Pairwise comparison of genetic differentiation ($F_{ST}$) among 15 *S. moorcroftiana* populations.**

|       | Pop1  | Pop2  | Pop3  | Pop4  | Pop5  | Pop6  | Pop7  | Pop8  | Pop9  | Pop10 | Pop11 | Pop12 | Pop13 | Pop14 |
|-------|-------|-------|-------|-------|-------|-------|-------|-------|-------|-------|-------|-------|-------|-------|
| Pop2  | 0.021 |       |       |       |       |       |       |       |       |       |       |       |       |       |
| Pop3  | 0.041 | 0.062 |       |       |       |       |       |       |       |       |       |       |       |       |
| Pop4  | 0.048 | 0.050 | 0.071 |       |       |       |       |       |       |       |       |       |       |       |
| Pop5  | 0.029 | 0.046 | 0.059 | 0.060 |       |       |       |       |       |       |       |       |       |       |
| Pop6  | 0.077 | 0.092 | 0.100 | 0.101 | 0.052 |       |       |       |       |       |       |       |       |       |
| Pop7  | 0.092 | 0.105 | 0.116 | 0.111 | 0.084 | 0.093 |       |       |       |       |       |       |       |       |
| Pop8  | 0.104 | 0.118 | 0.132 | 0.124 | 0.095 | 0.104 | 0.050 |       |       |       |       |       |       |       |
| Pop9  | 0.187 | 0.202 | 0.211 | 0.203 | 0.176 | 0.181 | 0.139 | 0.132 |       |       |       |       |       |       |
| Pop10 | 0.164 | 0.172 | 0.185 | 0.175 | 0.150 | 0.161 | 0.112 | 0.128 | 0.152 |       |       |       |       |       |
| Pop11 | 0.185 | 0.193 | 0.205 | 0.198 | 0.172 | 0.183 | 0.134 | 0.150 | 0.174 | 0.049 |       |       |       |       |
| Pop12 | 0.290 | 0.299 | 0.308 | 0.301 | 0.278 | 0.287 | 0.247 | 0.257 | 0.250 | 0.225 | 0.241 |       |       |       |
| Pop13 | 0.387 | 0.395 | 0.404 | 0.396 | 0.375 | 0.383 | 0.344 | 0.352 | 0.372 | 0.334 | 0.350 | 0.336 |       |       |
| Pop14 | 0.334 | 0.343 | 0.351 | 0.345 | 0.324 | 0.331 | 0.294 | 0.303 | 0.303 | 0.277 | 0.291 | 0.160 | 0.377 |       |
| Pop15 | 0.401 | 0.410 | 0.418 | 0.411 | 0.392 | 0.398 | 0.366 | 0.372 | 0.373 | 0.352 | 0.368 | 0.250 | 0.458 | 0.151 |

**Notes.**

PIC, polymorphism information content; Shannon, Shannon–Wiener index; PI, Nucleotide diversity; Ho, observed heterozygosity; He, expected heterozygosity.
Species-level estimates were computed treating the 15 populations as a whole.

was close to each other, and the genetic variation occurred within populations rather than among populations. The degree of genetic differentiation among the populations that were distributed close to one another, such as Pop 7 with Pop 8 and Pop 10 with Pop 11, was low. Pop 7 and Pop 8 generally exhibited a moderate level of genetic differentiation with the populations from Shigatse, including Pop 9, Pop 10, and Pop 11. There was a large genetic difference between Pop 13 and all other populations. This result is complementary to the results of the previous analysis.

## Gene flow among populations

Mutation, genetic drift due to a finite population size, and natural selection favoring adaptations to local environmental conditions will all lead to the genetic differentiation of local populations, and the movement of gametes, individuals, and even entire populations—collectively called gene flow—will oppose that differentiation (*Slatkin, 1987*). Slatkin indicated that gene flow may either constrain evolution by preventing adaptation to local conditions or promote evolution by spreading new genes and combinations of genes throughout a species' range. One reason for estimating *Nm* is that this combination of parameters indicates the relative strengths of gene flow and genetic drift. Genetic drift will result in substantial local differentiation if $Nm < 1$ but not if $Nm > 1$ (*Wright, 1951*). From the $F_{ST}$, we obtained the gene flow among the 15 populations (Table 4). The higher the gene flow, the lower the degree of genetic differentiation between populations. The gene flow among the Shigatse populations was generally at a high level (mean value for *Nm* is 4.668). All the *Nm* values among the 6 populations in Shigatse are greater than 1, especially between Pop 1 and Pop 2, where *Nm* peaked 11.601. In addition, gene flow exhibited higher levels among the populations distributed close to each other, such as Pop 7 with Pop 8 ($Nm = 4.750$) and Pop 10 with Pop 11 ($Nm = 4.826$). Pop 7 and Pop 8 (subgroup

**Table 4  Gene flow ($Nm$) among 15 *S. moorcroftiana* populations.**

|  | Pop1 | Pop2 | Pop3 | Pop4 | Pop5 | Pop6 | Pop7 | Pop8 | Pop9 | Pop10 | Pop11 | Pop12 | Pop13 | Pop14 |
|---|---|---|---|---|---|---|---|---|---|---|---|---|---|---|
| Pop2 | 11.601 | | | | | | | | | | | | | |
| Pop3 | 5.807 | 3.773 | | | | | | | | | | | | |
| Pop4 | 4.951 | 4.709 | 3.253 | | | | | | | | | | | |
| Pop5 | 8.228 | 5.226 | 4.003 | 3.943 | | | | | | | | | | |
| Pop6 | 3.013 | 2.458 | 2.260 | 2.231 | 4.568 | | | | | | | | | |
| Pop7 | 2.460 | 2.133 | 1.900 | 1.994 | 2.741 | 2.426 | | | | | | | | |
| Pop8 | 2.143 | 1.862 | 1.640 | 1.772 | 2.393 | 2.164 | 4.750 | | | | | | | |
| Pop9 | 1.085 | 0.988 | 0.935 | 0.980 | 1.171 | 1.129 | 1.544 | 1.639 | | | | | | |
| Pop10 | 1.278 | 1.203 | 1.102 | 1.179 | 1.417 | 1.299 | 1.981 | 1.708 | 1.399 | | | | | |
| Pop11 | 1.101 | 1.047 | 0.970 | 1.010 | 1.199 | 1.117 | 1.614 | 1.413 | 1.188 | 4.826 | | | | |
| Pop12 | 0.612 | 0.586 | 0.560 | 0.582 | 0.651 | 0.620 | 0.761 | 0.724 | 0.752 | 0.861 | 0.785 | | | |
| Pop13 | 0.395 | 0.382 | 0.368 | 0.381 | 0.417 | 0.403 | 0.477 | 0.461 | 0.422 | 0.498 | 0.464 | 0.494 | | |
| Pop14 | 0.498 | 0.478 | 0.463 | 0.474 | 0.522 | 0.505 | 0.600 | 0.576 | 0.576 | 0.653 | 0.611 | 1.314 | 0.414 | |
| Pop15 | 0.373 | 0.360 | 0.348 | 0.358 | 0.388 | 0.379 | 0.433 | 0.421 | 0.421 | 0.460 | 0.430 | 0.751 | 0.296 | 1.410 |

**Table 5  Estimates of genetic variability of 4 *S. moorcroftiana* subgroups.**

| Population | PIC | Shannon | PI | Tajimas'D | $Ho$ | $He$ |
|---|---|---|---|---|---|---|
| subgroup I | 0.189 | 0.359 | 0.00003204 | 1.857 | 0.2453 | 0.2745 |
| subgroup II | 0.192 | 0.363 | 0.00003293 | 1.148 | 0.2699 | 0.2834 |
| subgroup III | 0.195 | 0.370 | 0.00003314 | 1.253 | 0.2186 | 0.2704 |
| subgroup IV | 0.174 | 0.329 | 0.00003025 | 1.898 | 0.2438 | 0.3071 |

II) were geographically distributed between the Shigatse subgroup (subgroup I) and the Lhasa-Shannan mixed subgroup (including Pop 9, Pop 10, and Pop 11 in subgroup III), and there were extensive genetic exchanges between Pop 7, Pop 8 and the two subgroups ($Nm > 1$). The Nyingchi subgroup (including Pop 12, Pop 14, and Pop 15 in subgroup IV) and the Nyingchi Gongbu Jiangda subgroup (Pop 13 in subgroup V) had low levels of gene flow with the other populations.

## Analysis of genetic diversity and $Nm$ for the four main subgroups

According to the comprehensive results of the phylogenetic analysis, population structure analysis and PCA analysis, 15 natural populations could be clustered into five subgroups. In order to increase the sample size and verify the analysis results, we removed subgroup V (Pop 13), which has large genetic differences with all the other populations and analyzed the genetic diversity and $Nm$ of the following four main subgroups: subgroup I) Pop 1-6, subgroup II) Pop 7 and 8, subgroup III) Pop 9-11, and subgroup IV) Pop 12, 14 and 15. Among all the subgroups, subgroup II had higher genetic diversity (Table 5) and had high levels of gene flow with subgroup I ($Nm = 3.271$) and subgroup III ($Nm = 2.894$) (Table 6). Subgroup IV showed large genetic differentiation with the other subgroups ($Nm < 1$), especially with subgroup I and subgroup II. These results are consistent with our previous analysis.

**Table 6** Gene flow ( *Nm*) among 4 *S. moorcroftiana* subgroups.

|  | Subgroup I | Subgroup II | Subgroup III |
|---|---|---|---|
| Subgroup II | 3.271 | | |
| Subgroup III | 1.803 | 2.894 | |
| Subgroup IV | 0.633 | 0.744 | 0.918 |

## DISCUSSION

A study on the arenaceous adaptability of *S. moorcroftiana* (*Zhao, 1998*) revealed that its optimal habitats were terraces covered with sand, semi-motive dunes and the initial stages of fixed sand dunes. This species first invaded and became established in motive dunes by seed reproduction, then became abundant by establishing roots and finally declined owing to interspecific competition. A study on the soil seed bank of *S. moorcroftiana* (*Liu, Zhao & Li, 2002*) showed that 70% of all seeds were distributed on the surface. Seed dispersal was driven by wind, gravity (slope) and water flow, and the dispersal by water flow was closely related to the landforms and the carrying capacity of flowing water. Seeds dispersed by gravity or wind might have been carried away again by water. The regions alongside the Yarlung Zangbo River are covered with mobile dunes, and the degree of desertification in the Yarlung Zangbo River basin is as follows: Shigatse > Shannan > Lhasa > Nyingchi (*Xu, Li & Sun, 2006*). In this study, there were high levels of gene flow between populations in Shigatse, which inhibited the genetic differentiation between these populations. Most individuals of *S. moorcroftiana* grow in desertified areas (*Zou et al., 2002*), and the population density of *S. moorcroftiana* increased with increasing altitude (*Zhao, Zhang & Li, 2007*). Therefore, large seed banks are driven by ubiquitous quicksand, which may lead to this phenomenon. The middle reaches of the Yarlung Zangbo River form the center of social and economic development in Tibet and are severely affected by aeolian sands. Pop 7 and Pop 8 in the Lhasa River Basin, located between subgroup I and subgroup III, had higher levels of genetic variation among all populations and had a close genetic relationship between the two subgroups. This expansion direction is contrary to the direction of seed dispersal by water and gravity. A study of aeolian sandy land in the areas around the Lhasa Airport (*Li et al., 2010*) revealed the trends of wind direction changes in the areas around the Lhasa Airport from 1980–2006 (Fig. 6), showing that west wind (W), east wind (E) and east-north 22.5° wind (ENE) were frequent around Lhasa Airport. This pattern may explain why the genetic information of subgroup I and III was shared with subgroup II and why Pop 5 and Pop 6, at higher altitudes, shared genetic information with subgroup III. The research on genetic diversity of 10 populations of *S. moorcroftiana* near the Yarlung Zangbo River assessed by allozymes (*Zhao et al., 2003*) revealed that the population in the middle area (eastern Shigatse and western Shannan) harbored the majority of alleles and had high levels of genetic diversity, which is roughly consistent with our results. In the lower reaches of the Yarlung Zangbo River, Nyingchi has better climate and soil conditions and is rich in species and vegetation resources. *S. moorcroftiana*, which thrives on moving dunes, followed by semifixed dunes, had a declining trend after the sand was fixed (because other species colonized the fixed sand). In addition, there is a negative

effect of human pressure and habitat fragmentation. These factors may have resulted in lower population density in Nyingchi and lower gene flow with the populations in other areas. We found significant differences in the flowering period between different regions, which might also have some effect on the gene exchange between different populations. Pop 13 from Gongbu Jiangda county in Nyingchi is located on the eastern slope of Mount Mira in the upper reaches of the Niyang River, which has a temperate and moist climate and high vegetation coverage and species diversity. The environment hinders the spread of seeds and thus causes far genetic distances between Pop 13 and the other populations, except for Pop 12, which shared little genetic information with Pop 13 (potentially due to gravity). In the same basin as Pop 13, Pop 15 is located in the downstream of the Niyang River and is distantly genetically related to upstream Pop 13. The genetic distance between Pop 15 and Pop 13 was greater than the average distance between Pop 13 and the other populations. This finding also indirectly proves the suggested limitations of seed dispersal by water (*Liu, Zhao & Li, 2002*). Pop 9 is located in the upstream of the Lhasa River on the western slope of Mount Mira. There are clear differences in habitats between the western and eastern slopes of Mount Mira. The western slope has a cold and dry climate, and desertification is severe there. Pop 9 demonstrated extensive gene flow with the populations in adjacent regions, such as Pop 8, Pop 7, and Pop 10. These showed the importance of seed dispersal by gravity, sand and wind.

To explain these results obtained from the GBS SNP dataset, we considered the biotope and overall geographical environment and formulated 3 points on how *S. moorcroftiana* evolved. (1) *S. moorcroftiana* originated in high-altitude areas in the west and expanded to lower altitudes in the east via seed dispersal by gravity, wind, sand drifts and water flow. (2) Due to the effects of wind direction, the greatest number of seeds was dispersed into the central region mainly from the east, west, east-northeast and west-southwest, which was one of the greatest factors leading to the high genetic variation in Pop 7 and Pop 8, which are located in the central region. (3) The geographical environment, including the vegetation coverage and the degree of desertification degree, had a strong influence on the expansion of *S. moorcroftiana*. Pressure from humans may also have a great impact on the genetic characteristics of *S. moorcroftiana*.

Avise (*Avise & Hamrick, 1996*) pointed out a lack of concern for the genetic diversity of endemic species, and in recent years, the role of population genetics in plant conservation biology has received much attention (*Liu et al., 2006*; *Shao et al., 2009*; *Rayamajhi & Sharma, 2018*). Research on *S. moorcroftiana* has focused on its medicinal value (*Su et al., 2017*), alkaloid production (*Ma et al., 2018*), forage value, drought resistance (*Li et al., 2015*) and desertification control value (Zhao et al., 1998; *Zhao, Zhang & Li, 2007*). However, there have been few studies on *S. moorcroftiana* population genetics. The Tibetan flora is characterized as a young flora with high endemicity (*Wu, 1987*). A hypothesis was proposed in which *S. moorcroftiana* diverged from *S. davidii* and speciated (Wei & Shou, 1996). Wu hypothesized that the Tibetan flora originated mostly from the paleotropical Tertiary flora in the Hengduan Mountains by adapting to the cold and arid environments associated with the strong uplift of the QTP (*Wu, 1987*). The results of a study (*Cheng et al., 2017*) using the cpDNA and ITS of *S. moorcroftiana* and *S. davidii* to explore the

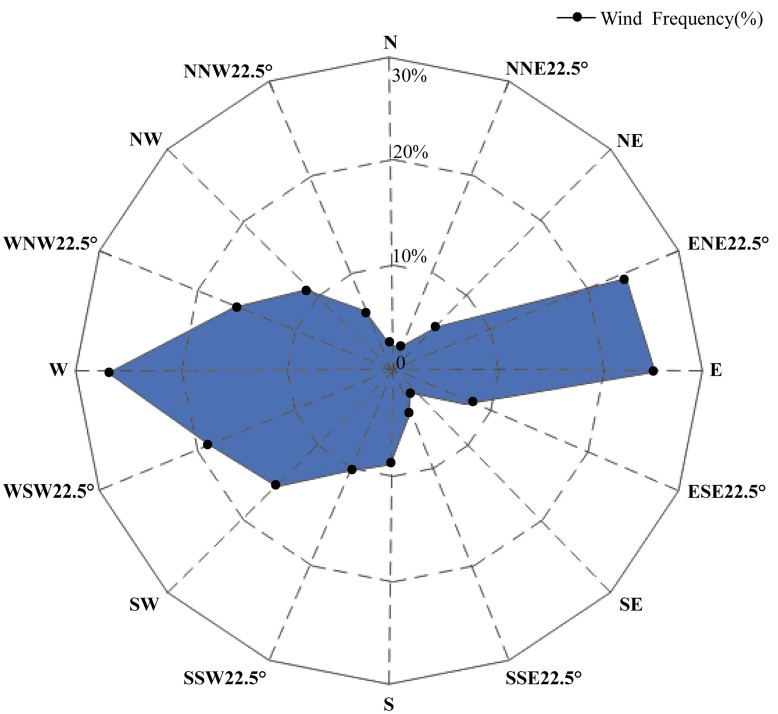

**Figure 6 Wind speed and direction change around the sampling area observed by Lhasa Airport from 1980–2006 (*Li et al., 2010*).** N, north; NNE, north-northeast; NE, northeast; ENE, east-northeast; E, east; ESE, east-southeast; SE, southeast; SSE, south-southeast; S, south; SSW, south-southwest; SW, southwest; WSW, west-southwest; W, west; WNW, west-northwest; NW, northwest; NNW, north-northwest.

relationship between the two species support these two hypotheses. This study of genetic diversity indicated that both total genetic diversity and within-population diversity in *S. moorcroftiana* are low, and suggested that *S. moorcroftiana* might have undergone serious bottleneck(s) and genetic drift, which might have been caused by the uplift of the QTP. In our study, the genetic diversity of *S. moorcroftiana* was also generally at a low level, and the relatively high level of genetic variation in the high-altitude areas, such as in all the Shigatse populations, may have resulted from *S. moorcroftiana's* adapting to harsh environments over a long period of time.

During the field investigation, we found abundant *S. moorcroftiana* seeds in the sand, and *S. moorcroftiana* has the reproductive characteristic of first invading and becoming established by seed reproduction and then becoming abundant via root turions. Seed banks have a direct effect on population dynamics (*Harper, 1977*), and 70% of the *S. moorcroftiana* seed bank in the middle reach of the Yarlung Zangbo River is distributed on the soil surface without litter (*Reichman, 1984*). The relatively closed habitat has a dense seed bank, showing a high seed yield of *S. moorcroftiana* seeds in the Yarlung Zangbo River (*Liu, Zhao & Li, 2002*). The seeds are dispersed by wind, gravity and water flow (*Fei & Ling, 1995*). The Yarlung Zangbo River originates in the southwestern part of Tibet and is located in the middle of the Himalayas. Due to the plateau's monsoons and subtropical westerly jet, the river valley exhibits a dry, cold and windy climate (*Dong et al., 1997*). The

altitude gradually decreases from the northwest to the southeast, and the orientation of the river valley is nearly parallel to the wind direction. In the winter and spring, the westerly jets are faster (maximum instantaneous wind speed: 35.2 m/s) and more frequent (*Dong, Li & Dong, 1999a*). These geographic and climatic conditions, such as wind direction, slope caused by the terrain, and flow direction, result in a dynamic environment and may lead to the expansion of *S. moorcroftiana* from high-altitude areas in the west to lower-altitude areas in the east. This expansion is consistent with the possible propagation direction of *S. moorcroftiana* proposed by *Liu et al. (2006)*, but the propagation mode was slightly different. Sand activity is the dominant factor affecting vegetation that grows in sand. In particular, the vegetation on semifixed and semimovable sand dunes is most seriously affected, and the species and coverage of vegetation can reflect the stability of sand sources. Vegetation succession proceeds based on sand activity (*Chang et al., 2006*), and strong winds are a major force affecting sand drift. When winds are strong, the movement of the sand dunes is very rapid (*Liu & Zhao, 2001*). The movement of seeds caused by sand drift and wind may have led to the high gene exchange flow of subgroup II with subgroup I and subgroup III, which are located in the central region. The resources with a high level of genetic diversity should be specifically protected and sustainably exploited to enable adaptation to and improvement of extreme environments. The location and environment of Gongbu Jiangda county may have caused the *S. moorcroftiana* individuals in this area to be relatively isolated, leading to a low level of gene flow and high levels of differentiation with other populations. Owing to the abundant rain and suitable temperature in Nyingchi, there is a high level of species diversity and a high ratio of vegetation coverage in this area. As a result of the low interspecific competitiveness of *S. moorcroftiana*, it is not the dominant species in Nyingchi. A lower population density and gene flow and human impacts may have led to the lower genetic diversity in Nyingchi. A low population genetic diversity is associated with a low ability to withstand threats (*Spielman, Brook & Frankham, 2004*). Thus, the protection of such populations should be strengthened.

The genetic characteristics and structure of *S. moorcroftiana* are related not only to climate and the geographical environment but also to insect-mediated pollen flow. Pollinator activity as well as dye or pollen dispersal are positively affected by plant population size, density or both (*Rossum & Triest, 2010*; *Nattero et al., 2011*), and most pollen is dispersed over short distances to the first few flowers visited and only occasionally over long ranges (leptokurtic distribution, (*Thomson & Plowright, 1980*; *Holmquist, Mitchell & Karron, 2012*). The habitat fragmentation caused by human activities such as overexploitation (*Young, Boyle & Brown, 1996*; *Dong & Li, 1999b*), grazing, cultivation and road construction also affects the genetic traits of *S. moorcroftiana*. *Llorens, Ayre & Whelan (2018)* indicated that anthropogenic fragmentation may not alter pre-existing patterns of genetic diversity and differentiation in perennial shrubs. Therefore, the main factors affecting the evolutionary history and genetic relationship of *S. moorcroftiana* are climate and the geographical environment. *S. moorcroftiana* is of great significance to Tibet and other desertified areas, and artificial planting and protection of this species should thus be promoted.

## CONCLUSIONS

Through previous studies, we knew that *S. moorcroftiana* diverged from *S. davidii* and speciated in the uplift of the QTP. The driving forces of *S. moorcroftiana* seed dispersal were wind, gravity (slope) and water flow. Furthermore, the unique geographical environment of the QTP caused *S. moorcroftiana* to expand from the western high-altitude to the eastern low- altitude areas and from the south and north areas to the Yarlung Zangbo River bank. Our research found that there was a high-level of east–west differentiation in *S. moorcroftiana* populations in the Yarlung Zangbo River basin due to the different geographical environment. The aeolian desertification of the Yarlung Zangbo River Basin is severe, and seed dispersal driven by wind and sand flow is one of the main factors for the large gene flow between the central groups and eastern and western groups. Thus, seed dispersal caused by gravity, wind, sand flow and geographical distance are the main driving forces contributing to the diffusion pattern of *S. moorcroftiana* populations. Although *S. moorcroftiana* seeds expansion can be driven by water flow, water flow is limited because it is closely related to the landforms and the carrying capacity of flowing water. The barriers between mountains and the competition between *S. moorcroftiana* and other species are disadvantages for the expansion of *S. moorcroftiana*. Considering that *S. moorcroftiana* is of great value in controlling aeolian desertification in Tibet (especially for motive dune and semi-motive dune lands), we should spare no effort to protect and utilize genetic diversity and germplasm resources.

## ACKNOWLEDGEMENTS

We are grateful to Minghao Zhang for his guidance on software usage and data analysis.

### Funding

This research was funded by the Forestry Industry Research Special Funds for Public Welfare Projects (No. 201504109). The funders had no role in study design, data collection and analysis, decision to publish, or preparation of the manuscript.

### Grant Disclosures

The following grant information was disclosed by the authors:
Forestry Industry Research Special Funds for Public Welfare Projects:  201504109.

### Competing Interests

The authors declare there are no competing of interests.

### Author Contributions

- Ying Liu conceived and designed the experiments, performed the experiments, analyzed the data, prepared figures and/or tables, authored or reviewed drafts of the paper, and approved the final draft.

- Fei Yi and Ciren Pubu performed the experiments, authored or reviewed drafts of the paper, and approved the final draft.
- Guijuan Yang and Junhui Wang conceived and designed the experiments, authored or reviewed drafts of the paper, and approved the final draft.
- Yuting Wang performed the experiments, prepared figures and/or tables, and approved the final draft.
- Runhua He and Nan Lu analyzed the data, prepared figures and/or tables, and approved the final draft.
- Yao Xiao analyzed the data, authored or reviewed drafts of the paper, and approved the final draft.
- Junchen Wang performed the experiments, analyzed the data, prepared figures and/or tables, authored or reviewed drafts of the paper, and approved the final draft.
- Wenjun Ma conceived and designed the experiments, prepared figures and/or tables, authored or reviewed drafts of the paper, and approved the final draft.

## Data Availability

The sequences of Sophora moorcroftiana and Sophora davidii are available at SRA: SRP213569.

## Supplemental Information

Supplemental information for this article can be found online at http://dx.doi.org/10.7717/peerj.9609#supplemental-information.

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
