# Peer review of "Geographic population genetic structure and diversity of Sophora moorcroftiana based on genotyping-by-sequencing (GBS)"

_PeerJ, doi:10.7717/peerj.9609_

## Round 0.1 · original submission · Major Revisions

Dear authors, please make sure that you provide careful responses to all issues identified by the reviewers. It is important that the link to the raw data is valid, raw data should be made available in a public repository such as NCBI SRA to ensure reproducibility of the analysis. Description of the experimental methodology should be clear and comprehensive, please, present all parameters. It has been noted that GBS does not allow to reconstruct the phylogeny - you need to revise your approach to take this into account. It is important to present statistical summaries to support your conclusions. Overall, your manuscript addresses an interesting problem. I encourage you to address the concerns of the reviewers and improve your manuscript.

Reviewer 1 ·

Basic reporting

A. Important concerns

1. The introduction section lacks background information on the use of molecular markers on Sophorora moorcroftiana or related species.

2. Lines 100-108, This sounds more like your methodology! Delete it.


3. The raw data is not available online through the link provide by the authors. I suggest that you check it.

B. Minor concerns

1. I suggest that you add geography location on ADMIXTURE Figure 3

2. Provide legend color for ancestors according to ADMIXTURE Figure 3

3. Lines 121,124, avoid to start a sentence with abbreviation and number.

4. English language is understandable but I am not an English native speaker and I cannot judge if it needs correction.

Experimental design

A. Important concerns

1. Provide a climate description of the sampling locations

2. Line 145, why did you combine the two paired end reads?

B. Minor concerns

1. How did you choose the sampling locations?

2. Line 166, It is GCTA-PCA and provide reference as recommended here: http://cnsgenomics.com/software/gcta/#PCA

3. Lines 154 and 162, rewrite these parameters without abbreviations and by removing the two dashes.

4. Line 144, provide your python script as additional file.

Validity of the findings

A. Important concerns

1. Phylogenetic analysis

With GBS data you cannot construct species trees or gene trees because phylogenetics models have not been developed to take into account for example the amount of hybridization, incomplete lineage found at the intraspecific level. I suggest that you read this discussion (https://groups.google.com/forum/#!topic/raxml/1oSo4__nbms) where the author of RaxML is embarrassed with the use of RaxML for population genetics sampling. Therefore NJ-Tree is more powerful compared to RaxML.

2. Denovo assembly and snps calling on stacks

There are many missing information regarding the parameters and output results of stacks (SNPs data analysis section lines 146-176). First of all, the authors didn’t provide the version of stacks used and the parameters for denovo assembly using ustacks, cstacks, etc are missing. Secondly, I don’t understand why you used other software (PGDSpider_cli, VCFtools) to convert vcf format to phylogenetic format and for Hw, Ho, He, Fst calculations whereas all that can be provided by stacks (http://catchenlab.life.illinois.edu/stacks/comp/populations.php) ? Finally, the authors didn’t provide statistics for denovo assembly (Number of loci, number of snp, the depth,..). I suggest that you add statistics summary results of stacks as additional files.

3. Ploidy level

What is the ploidy level of the samples used in this study? Did you estimate the ploidy level of the samples used in this study?

Additional comments

The manuscript submitted is very useful for Sophorora moorcroftiana conservation. However, the authors omitted information about data analysis and without all these information this study seems not repeatable and I suggest that the author use the NJ-tree instead of RaxML for the phylogenetic tree. So, if the authors fully address all the points that I raised and provide the missing elements regarding data analysis, I encourage the authors to submit the revised version of the manuscript.

Reviewer 2 ·

Basic reporting

Major comments:
1. According to the journal policies, raw data should be made available in a public repository such as NCBI SRA to ensure reproducibility of the analysis. I think in this case it may be worth to also make available the VCF file with raw SNP genotype calls provided by Stacks.

Minor comments
1. Some basic statistics should be reported as supplementary material, including the total reads per sample, the number of raw and filtered SNPs, distributions of minor allele frequency and observed heterozygosity both overall and within populations, percentage of SNPs genotyped per individual and percentage of heterozygous genotype calls per individual. All this information would allow to make much more informed filtering decisions.

2. In the admixture analysis (Figure 3) it would be much more informative if all k values from K=2 to at least K=7 are presented. I personally do not find useful the test to identify an optimum K value. On the other hand, presenting the progression of the clustering across k values provides information on the importance and stability of each cluster. I would make figure 3A supplementary to accommodate the results of the clustering with other K values. Please also organize the populations from 1 to 15. I do not see any rationale for the current organization of populations in figure 3.

3. About the structure of the document, I would move part of the current text of the discussion to the appropriate results section to provide readers with a better context of each result. The current results text looks just as a statement of the observed numbers and then in the discussion (especially the first three paragraphs) I had to keep looking back at the results. I think this change will also help the discussion to be more focused on the overall description of the evolution of the species and the importance of the obtained results. I would also remove the conclusions section because it just looks like a summary of the discussion.

4. I do not think it is worth to make and present separate regressions of diversity indexes with longitude and altitude given that these two values have a negative correlation of 0.93. The linear regression between longitude and altitude could be presented in one side of figure 1 and then only one of the variables could be related with diversity indexes.

Experimental design

Major comments:
1. The average observed heterozygosities are surprisingly high (far above 0.5 for all populations). This indicates that there should be an important number of variants for which all samples within populations are heterozygous. Biologically, the only explanation for this phenomenon would be high ploidy. Although this does not seem to be the case for this species, if this is the case, then variant genotyping should be modified to take into account this phenomenon and estimate allele dosages per genotype. More likely, heterozygosity rates could be inflated by false heterozygous calls produced by the stacks variant caller and lack of filtering of repetitive regions. Because the analysis is made de-novo (it looks like a reference genome for the species is not available), the repetitive regions are not known. One quick alternative to filter these variants is to perform a relative distance based filtering, for example keeping only one variant for each stack. An explicit filtering of maximum observed heterozygosity across the 225 individuals could also be performed to remove variants with abnormally high heterozygosity values. Please calculate the distributions of global expected and observed heterozygosity across the entire population. Given the observed population structure, observed heterozygosity should be generally smaller than expected heterozygosity due to the Wahlund effect.

2. The number of sampled individuals per population is a bit low to support most of the results presented in the manuscript. Moreover, if a maximum of 80% missing data per SNP is being allowed (as indicated in the methods section), the number of genotyped individuals per population can be too low to perform accurate estimations of population level statistics. For a given SNP, it could even be the case that some population have only one or even zero individuals genotyped (this could also explain the inflated observed heterozygosities). This issue can severely skew all results presented in the manuscript. Filtering for population structure analysis should be generally conservative. Normally, I recommend to have less than 5% of missing data overall.

3. If the population structure remains stable after improving the filtering process, my recommendation would be to make correlations of genetic diversity with longitude and latitude, and the analysis of gene flow considering only 4 groups: 1) populations 1-6, 2) populations 7 and 8, 3) populations 9,10 and 11, and 4) populations 12, 14 and 15. This way, each group would have enough genotyped individuals for accurate estimation of population level statistics. Instead of a linear regression, simple tests of difference between means could be used to identify significant differences in diversity values between these groups.

Validity of the findings

Minor comments:
1. I do not understand why throughout the manuscript populations 7 and 8 are split across two groups. In all clustering analyses these two populations appear as clustering together.

Additional comments

The manuscript describes a population genetic analysis of the perennial leguminous shrub Sophora moorcroftiana in 15 different high altitude locations across the Yarlung Zangbo River and three of its affluents. The study is interesting, relevant for population genetics and generally well conducted. The manuscript is clearly written and has good English quality. Major and minor comments are included in the relevant sections above.

Reviewer 3 ·

Basic reporting

It is an interesting study. I think the topic of the paper is of interest to the audience of the journal. However, the manuscript in its present form has several weaknesses.
1. Τhe authors have to improve the language. The manuscript has to be edited by a native English speaker or a professional.
2. The abstract is very general especial the results.
3. A part of the introduction is very general and it is not related to the present study.
4. The authors have to rewrite the conclusions which more or less are replication of the results.

Experimental design

The research is well orginised. However, the authors have to give more information about the sampling sites.

Validity of the findings

no comment

Additional comments

Some of my suggestions are indicated in accompanying document

Annotated reviews are not available for download in order to protect the identity of reviewers who chose to remain anonymous.

---

## Round 0.2 · Major Revisions

We have received informative comments from the reviewers. I understand that the person who did the bioinformatics analysis for your paper is no longer available. Since the concerns raised about the validity of the data analysis are serious, calculations of diversity statistics need refinement, you may need to involve an additional bioinformatician to help you. The filtering of SNP based on various quality parameters is a long and painful process. However, without this step, the results may be grossly misleading. Please take the comments by the reviewers extremely seriously and provide a point-by-point constructive response..

Reviewer 2 ·

Basic reporting

No further comments

Experimental design

Unfortunately, the authors have not seem to make any change relative to my major concern about the calculations of diversity statistics. Although the authors claim that they changed the parameter “-max-missing” of VCFTools from 80% to 60%, Table 2 is exactly the same as in the original version. This indicates that no alternative filtering was really tried to solve the technical issue reflected in high heterozygosities. Also, the authors seem to misunderstand this parameter. In the methods thext they write “the SNP was called at least 60% of individuals” but according to the documentation of VCFTools it would be exactly the opposite. The consequence of this is that there could be an important number of SNPs with missing data in every individual of the population.

Based on the new supplementary table with SNP statistics per sample, I infer that the total number of SNPs in the VCF used to calculate the diversity statistics is about 400.000. In all GBS experiments that I have analyzed, this corresponds to the number of raw variants and normally contains a large number of false positives, either because each false positive heterozygous call in at least one sample becomes an new false positive SNP or (most likely in this case) repetitive regions contribute a lot of SNPs that are heterozygous in most of the samples and skew population statistics. The percentage of heterozygous SNPs per sample (around 70%) seems to confirm this. Unfortunately, the distribution of minor allele frequency and observed heterozygosity that I requested for the entire set of samples is not provided. These distributions are needed to see the trend in the dataset of SNPs and identify possible technical issues.

To make it clearer, this is more or less the filtering procedure that the authors should apply to the data:

1. Minimum genotype quality 30
2. Distance between variants of 300bp. This is to keep only one SNP per tag and indierctly remove some of the issues with repetitive regions.
2. At least 80% of the samples genotyped (in VCFTools it would be -max-missing 20%)
3. MAF above 0.05

Then, distributions (not only averages) of MAF and observed heterozygosity should be provided for both the entire population of 225 samples and the 4 main groups: (1) populations 1-6, 2) populations 7 and 8, 3) populations 9,10 and 11, and 4) populations 12, 14 and 15). If the authors insist on calculating and making inferences on diversity statistics for each of the 15 populations, then they need to perform additional filtering to keep only SNPs genotyped in 14 out of 15 individuals of each population to try to make inferences based on somewhat reliable numbers.

The authors mention in the rebuttal that “the amount of data in the physical file was too large. If we adjust the parameters too small, it will make the calculation too difficult”. This does not make any sense. A VCF with 400.000 SNPs genotyped in 225 individuals should have for sure less than 100Gb. The procedure described above can be done in a normal laptop and there are plenty of open source software tools useful to perform these filtering steps.

The authors also write in the rebuttal that “However, the previous analysis was done with the help of others. Even though I have tried to perfect it , I am so sorry that I couldn’t finish it within the bound of my abilities”. Whoever performed the bioinformatic analysis in the first place should be able to perform another round of analysis, especially taking into account that important technical issues need to be solved.

As minor comment, the text should not contain qualitative statements about population statistics such as “the Ho is less valuable because it is affected by inbreeding and other evolutionary processes” or “Fst is an excellent measure of genetic differentiation”. Population statistics are just different sources of information of diversity and they just need to be properly calculated to make accurate inferences. For example, the claim that “the populations conforms to Hardy-Weinberg Equilibrium” would not even make sense if the observed heterozygosity is not calculated because HWE is tested as the difference between observed and expected heterozygosity. Also, from a strictly statistical point of view, HWE is not really accepted because that is equivalent to accept the null hypothesis.

Validity of the findings

No further comments

Additional comments

Although I generally like the study and it is clear to me that the authors know the ecology of the species and the sampling is appropriate to draw some of the conclusions, I can not accept the manuscript if a proper bioinformatics analysis is not performed. If the authors are currently lacking bioinformatics support, they should collaborate with a bioinformatician and make proper acknowledgement of his/her contribution to be able to bring this study to a good end.

Reviewer 3 ·

Basic reporting

no comment

Experimental design

no comments

Validity of the findings

no comments

Additional comments

The manuscript has been much improved by the authors. I have only few minor suggestions in the attached file

Annotated reviews are not available for download in order to protect the identity of reviewers who chose to remain anonymous.

---

## Round 0.3 · Major Revisions

Please consider the comments of the reviewers very carefully. There was a suggestion that your dataset has not been LD-pruned. If it was, the procedure should be described. If not, please follow the suggestions and select the unlinked set of SNPs.

Reviewer 1 ·

Basic reporting

A. Important concerns

There are some typo problems (e.g. line 127 ultilized). Although the English language is understanding; you should give your manuscript to a native English speaker colleague for correction.

B. Minor concerns

Snps data analysis subsection line 157, It is not a python script but a shell command called “cat” that you used to combine the paired-end reads.

Experimental design

No comment

Validity of the findings

1. Admixture and structure analysis require unlinked SNPs data, you reported 400. 000 snps with not mention if they are linked or unlinked. I do think there linked snps among and therefore the analysis is biased. Furthermore, it seems like you don’t understand stacks program and parameters that is the reason why you used vcftools for filter purpose whereas everything is done with stacks, you should carefully read stacks manual (http://catchenlab.life.illinois.edu/stacks/manual/). However, here are my recommendations of some stacks parameters that you should pay attention for population structure analysis (http://catchenlab.life.illinois.edu/stacks/comp/populations.php):

-- write-single-snp: restrict data analysis to only the first SNP per locus and therefore snps are not linked.
You should try as well --write-random-snp (restrict data analysis to one random SNP per locus) to see this affect the result.
-p minimum number of populations a locus must be present in to process a locus
-R: minimum percentage of individuals across populations required to process a locus
-r minimum percentage of individuals in a population required to process a locus for that population
--structure: output results in Structure format.
As well you can generate many output files see the file output options ((http://catchenlab.life.illinois.edu/stacks/comp/populations.php)

To choose the best stacks parameters you should read this papers: https://besjournals.onlinelibrary.wiley.com/doi/full/10.1111/2041-210X.12775

2. Although you used Sophora davidii as outgroup species to avoid ascertainment bias this may not cluster well individuals of the targeted species if the outgroup species are genetically distant of S. moorcroftiana. Therefore, I suggest that you build an unrooted tree as well.

Additional comments

This is a great and pionner article for the S. moorcroftiana species. However, I warmly suggest that you should take carefully the recommendation regarding the data analysis (population structure and phylogenetic analysis) which is the weakness part of this manuscript.

Reviewer 2 ·

Basic reporting

I have no major comments. I found a few typos and small fixes that the author can make:

Line 96: Scrub
Line 121: Ultilized
Line 147: The command "cat" is not really a Python script. You can just say "bash command cat". About the script, cat itself would not join two compressed files. The command is either:
cat <SAMPLE>_1.fastq <SAMPLE>_2.fastq > <SAMPLE>.fastq

or

zcat <SAMPLE>_1.fastq.gz <SAMPLE>_2.fastq.gz | gzip -c > <SAMPLE>.fastq.gz

Line 192: Populaions

Experimental design

This time the authors addressed my previous comments and updated the analysis. The analysis looks fine now. I do not have any further comments

Validity of the findings

No comment

Additional comments

I am glad to see that the authors could improve the analysis as suggested and that it was helpful.

---

## Round 0.4 · accepted · Accept

Thank you for implementing all the proposed changes.

Reviewer 1 ·

Basic reporting

You should rename the additional file python to shell or bash script.

Experimental design

The authors addressed all my concerns and the results look more accurate now.

Validity of the findings

No comment

Additional comments

You improved a lot your manuscript and it looks fine now.

Reviewer 2 ·

Basic reporting

I have no further comments.

Experimental design

I have no further comments.

Validity of the findings

I have no further comments.

Additional comments

I have no further comments. I am glad to see that the manuscript went through editorial review to improve the english quality of the paper

Reviewer 3 ·

Basic reporting

no comment

Experimental design

no comment

Validity of the findings

no comment

Additional comments

Authors' responses and changes in the manuscript are satisfactory and I have no further comments. In my opinion, the manuscript can be accepted